# Effects of Cysteine and Inorganic Sulfur Applications at Different Growth Stages on Grain Protein and End-Use Quality in Wheat

**DOI:** 10.3390/foods11203252

**Published:** 2022-10-18

**Authors:** Jian Cai, Fujuan Zang, Liang Xin, Qin Zhou, Xiao Wang, Yingxin Zhong, Mei Huang, Tingbo Dai, Dong Jiang

**Affiliations:** College of Agriculture, Nanjing Agricultural University, No. 1 Weigang Road, Nanjing 210095, China

**Keywords:** wheat, sulfur and cysteine, different application stage, protein quality, flour quality

## Abstract

The aim of this study was to test the significant effects of inorganic sulfur and cysteine on grain protein and flour quality in wheat and to provide a theoretical basis of wheat cultivation techniques with high yield and quality. In the field experiment, a winter wheat cultivar, Yangmai 16, was used, and five treatments were established, i.e., S_0_ (no sulfur fertilizer application during the whole wheat growth period), S(B)_60_ (60 kg ha^−^^1^ inorganic sulfur fertilizer was applied as the basal fertilizer), Cys(B)_60_ (60 kg ha^−^^1^ cysteine sulfur fertilizer was applied as the basal fertilizer), S(J)_60_ (60 kg ha^−^^1^ inorganic sulfur fertilizer was applied as the jointing fertilizer), and Cys(J)_60_ (60 kg ha^−^^1^ cysteine sulfur fertilizer was applied as the jointing fertilizer). The fertilizer application at jointing stage showed a better influence than basal fertilizer application on protein quality; for the content of albumin, gliadin, and high molecular weight glutenin (HMW-GS), Cys(J)_60_ was the best among these treatments. An increase of 7.9%, 24.4%, 43.5%, 22.7% and 36.4% was found in grain yield, glutenin content, glutenin macro-polymer (GMP), low molecular weight glutenin (LMW-GS), and S content under Cys(J)_60_, in relation to the control, respectively. A similar trend was found in the end-use quality, as exemplified by an increase of 38.6%, 10.9%, 60.5%, and 109.8% in wet gluten content, dry gluten content, sedimentation volume, and bread-specific volume, respectively; a decrease of 69.3% and 69.1% in bread hardness and bread chewiness was found under Cys(J)_60_. In terms of application period, topdressing at jointing stage is compared with base fertilizer, the sulfur fertilizer application at jointing stage showed larger effects on grain protein and flour quality, from the different types of sulfur fertilizer, the application of cysteine performed better than the use of inorganic sulfur. The Cys(J)_60_ exhibited the best effects on protein and flour quality. It was suggested that sufficient sulfur application at jointing stage has the potential to enhance the grain protein and flour quality.

## 1. Introduction

Sulfur (S) is an essential mineral element for crops. Soil sulfur deficiency has increased in prevalence around the world as a result of the widespread use of high-purity nitrogen fertilizer and the decrease in the use of conventional organic fertilizer [1]. In China, arable land with a sulfur deficiency covers around 4 million hectares, or 30% of the total area of arable land. Additionally, 4 million hectares of potential sulfur deficient area poses a significant threat to wheat output [2].

Sulfur is absorbed and assimilated into plants to synthesize cysteine, which is involved in complex metabolic processes and plays an important role in the formation of grain protein [3,4]. It is also crucial for disulfide bond formation. The number of free sulfhydryl groups determines the formation of high molecular weight glutenin subunits (HMW-GS), low molecular weight glutenin subunits (LMW-GS), and the glutenin macropolymers (GMP) [5,6,7]. The study of Yoshino et al. [8] confirmed that the amount of disulfide bonds in flour also dropped along with the quantity of sulfur in wheat grains, which in turn affected the viscosity of flour. The quality of flour can be modified by sulfur fertilizer. Adding sulfur fertilizer can increase wet gluten content and flour settling value of wheat, prolong dough formation time, and decrease dough stability time and tensile resistance [9,10]. The sulfur fertilizer can not only improve the wet gluten content of wheat, but also improve the volume of bread, specific volume, and dough stability [2]. Previous studies suggested the addition of S increased loaf volume significantly at two sites where grain S concentration was also significantly increased and grain N:S ratio decreased. Application of the extra 50 kg ha^−^^1^ N increased grain protein concentration but did not increase loaf volume at any of the sites [11]. The effect of sulfur on improving quality varied by species and location [12]. The effect of sulfur on the synthesis of HMW-GS and LMW-GS and GMP polymerization is mainly achieved by regulating the synthesis of sulfur-containing amino acid cysteine [13]. Due to the sulfhydryl group on cysteine, two cysteines can combine to create a disulfide bond, which is crucial for the integrity of protein structure [14]. Therefore, it is of great significance to clarify the regulation effect of sulfur and cysteine fertilizer on wheat grain protein, GMP, HMW-GS, LMW-GS, and disulfide bonds, and to explore the mechanism of sulfur and cysteine fertilizer on wheat protein quality.

There is a great difference in the demand for fertilizer in different growth stages of wheat. The majority of the nutrients needed for wheat growth until the three-leaf stage originate from the endosperm, while the requirement for fertilizer increases after the tillering stage [15]. Raffan et al. [16] discovered that sulfur absorption was faster in wheat throughout the middle development stage, with the peak absorption rate occurring between jointing and booting stage. Yildiz et al. [17] reported that with a suitable sulfur supply, sulfur absorption rises steadily after returning to the green stage and peaked during the jointing to heading stage in wheat. However, it is unclear how different sulfur fertilizers used at various application stages may affect wheat quality regulation at present.

In this paper, sulfur fertilizer and cysteine fertilizer were applied before sowing or at the jointing stage, respectively, under field conditions. The contents of protein components, glutenin subunits, glutenin macropolymer, and bread baking quality were evaluated. We aimed to reveal the effects of sulfur or cysteine fertilizer application on protein quality and on bread-making quality. The results should help to provide a novel view for improving quality of wheat flour by sulfur and cysteine fertilization.

## 2. Materials and Methods

### 2.1. Experimental Design and Sampling

This experiment was conducted in Tangquan Farm (32°05′48.70″ N, 118°27′40.03″ E), Nanjing City, Jiangsu Province, from 2014 to 2015. The test material was medium gluten wheat variety Yangmai 16, and the previous crop was rice. The sowing time was 17 November 2014, and the harvest time was 28 May 2015. The nutrient contents of soil in the experimental field were as follows: total nitrogen 0.113%, alkali-hydrolyzable nitrogen 57 mg kg^−1^, available phosphorus 40.3 mg kg^−1^, available potassium 288.07 mg kg^−1^, organic matter 1.57%, and available sulfur 16.38 mg kg^−1^.

The field experiments were laid out in a single-factor completely randomized design with three replicates for each treatment. Five different treatments were applied: no sulfur fertilizer treatment (S_0_), 60 kg ha^−1^ inorganic sulfur fertilizer was applied as the basal fertilizer (S(B)_60_), 60 kg ha^−1^ cysteine sulfur fertilizer was applied as the basal fertilizer (Cys(B)_60_), 60 kg ha^−1^ inorganic sulfur fertilizer was applied as the jointing fertilizer (S(J)_60_), and 60 kg ha^−1^ cysteine sulfur fertilizer was applied as the jointing fertilizer (Cys(J)_60_). The plot area was 3.2 m × 3 m = 9.6 m^2^ (15 rows and 20 cm between rows), and there were 15 plots in total. After sowing, the basic seedlings were 240 × 10^4^ ha^−1^. The N fertilizer was applied before planting as the basal N and applied at jointing stage as the top-dressed N, at the rates of 120 kg urea ha^−1^, respectively. The P (applied as calcium superphosphate) and K (applied as potassium chloride) were mixed into the soil before planting at the rates of 120 kg P_2_O_5_ ha^−1^ and 120 kg K_2_O ha^−1^, respectively. Field management refers to local conventional cultivation techniques.

At maturity, the grains were harvested according to the plot, and the impurities were removed. The grains were stored at room temperature for one month. The grains were ground with FJ-1 grain experimental grinder (Zhuozhou Grain and Oil Machinery Factory, Hebei, China), and then screened with YFS-08 powder sieve (Zhongtai Technology, Henan, China), and the samples were used for the determination of relevant quality indicators.

### 2.2. Contents of Protein and Protein Components

According to the solubility of proteins in various solvents, four protein components, namely albumin, globulin, gliadin, and glutenin, were extracted in sequence according to American Association of Cereal Chemists 2000 (AACC 2000) [18]. Extraction of albumin was performed as follows: 1 g of flour was weighed and placed in a test tube, and 10 mL of distilled water was added. Then, the mixture was stirred for 30 min. The supernatant was centrifuged at 4000 *g* for 5 min and transferred to a sterilizing tube. We repeated the operation 4 times and steam-dried. The extracts of globulin, gliadin, and glutelin were obtained in 100 g L^−1^ NaCl, 70% (*w*/*v*) ethanol, and 2 g L^−1^ NaOH, respectively. The grain N content was determined using the semi-micro Kjeldahl method, which was multiplied by the coefficient of 5.7 to get the content of protein and protein components [19].

### 2.3. GMP Content

The determination of GMP content was determined according to the method of Don et al. [20]. The flour sample (50 mg) was suspended in 1 mL of 1.5% sodium dodecyl sulfate (SDS) solution and centrifuged at 15,500 *g* for 30 min at 20 °C. The supernatant was decanted. The tubes with flour sample were rinsed 2 times with 2 mL SDS (1.5%) solution and were drained upside-down. Next, 2 mL 0.2% sodium hydroxide solution was added, swirled, and shaken for 30 min, then 3 mL of biuret reagent was added, shaken, and rested for 30 min. The nitrogen content in the sediment measured with the biuret reagent [21] was taken as the GMP content.

### 2.4. GMP Particle Size

The extraction method of GMP was slightly modified from the method of Don et al. [20] and optimization according to Weegels et al. [22]. Weighed 1.5 g defatted flour was added into a 50 mL centrifuge tube, after which 30 mL 1.5% SDS solution was slowly added while vortexing. After sufficient vortexing, centrifugation was applied at 75,500 *g* at 25 °C for 30 min. The supernatant was discarded and was carefully extracted from the gelatinous transparent substance in the upper and middle layers of the centrifuge tube, then transferred to the standby 50 mL centrifuge tube. Then, 1.5% SDS solution was further added, and further vortexing was applied to thoroughly disperse the gel. Then, the particle count and size analyzer (Elzone II 5390, Microeritics, USA) manufactured was used for analysis.

### 2.5. Quantifications of HMW-GS and LMW-GS

The determination method of HMW-GS and LMW-GS was slightly modified according to Ji et al. [23] and Tatham et al. [24]. To weigh 0.08 g flour, 1.5 mL n-propanol (solution A) was added, and placed in a water bath at 65 °C for 30 min. Centrifugation was performed at 8900 *g* for 10 min. The supernatant was discarded, and the process was repeated three times. Solution B containing 1% dithiothreitol (DTT) (50% n-propanol, 0.08 M Tris-HCl, pH8.0) was added to the precipitate, which was incubated at 65 °C for 30 min, and then incubated at 65 °C for 15 min with 0.4 mL of solution B containing 1.4% 4-vinylpyridine (4-VP) for 5 min. After shaking, centrifugation at 8900 *g* was performed for 5 min. The supernatant was moved into a clean test tube. The sample was mixed with an equal volume of buffer solution (0.4% sodium dodecyl sulfate (SDS), 40% glycerol, 50 μM Tris-HCl (pH 6.8), 2% mercaptoethanol, 0.01% bromophenol blue), shaken well, then bathed in boiling water for 5 min, slightly cooled, and centrifuged at 4000 *g* for 10 min. The supernatant was extracted and separated for further SDS-PAGE analysis.

Sodium dodecyl sulfate-polyacrylamide gel electrophoresis (SDS-PAGE):

The concentration of separation gel was 13% Acrylamide:Bisacrylamide (Acr:Bis = 29:1), 0.375 M Tris-HCl (pH = 8.8), 0.1% SDS. The concentration of concentrated gel was 4% (Acr: Bis = 37.5:1), 0.125 M Tris-HC1 (pH 6.8), and 0.1% SDS. Electrode buffer was 0.025 M Tris, 0.19 M Glycine, 0.1% SDS. The current of each concentrated gel was 15mA, and for the separation gel it was 20mA. The total time of electrophoresis was about 10 h.

After the electrophoresis, the gel was washed in 12% trichloroacetic acid for 10 min, then washed with distilled water and dyed overnight. The dye solution was 40% ethanol, 7% acetic acid, and 0.1% Coomassie brilliant blue (R-250). Subsequently, gels were placed in a decolorizing solution (25% ethanol, 8% acetic acid) to clear the background, then scanned. Quantitative analysis was done with Quantity One (LICHEN, Changsha, China).

### 2.6. Contents of Nitrogen and Sulfur

The content of nitrogen was determined by semi-micro Kjeldahl method, and the total sulfur was digested by HNO_3_-HCl-HClO and determined by turbidimetric method [25].

### 2.7. Gluten Quality

A near-infrared reflectance instrument (7250 NIR, Perten Instruments, Stockholm, Sweden) was used to estimate the flour sedimentation value and other related quality indexes.

Gluten value referred to the determination of wet gluten in wheat flour by mechanical means according to Chinese National Standard GB/T 5506.2-2008 [26]. The flour was machine-washed with 2200 gluten instrument (MJ-III, Hangzhou, China), and after centrifugation, the sifted and unsifted gluten were weighed separately and were used for calculating the gluten index. After mixing, the sifted and unsifted gluten were dried and weighed for the calculation of dry gluten content.

### 2.8. Bread Baking Quality

Breads were prepared using the straight-dough method according to Chinese National Standard GB/T 14611-2008 (2008) [27].

The formula contained 100 g flour, 6 g sugar, 4 g skim milk powder, 3 g shortening, 1.8 g instant dry yeast, 1.5 g salt, 0.2 g wheat malt flour, and 4 mg ascorbic acid. Salt and sugar were dispersed in water. The mixture solution of salt and sugar was added to the premixed dry ingredients. Cohesive dough was prepared by hand. Amount of water added, mixing, and proof time were adjusted according to the performance of the dough. The dough was hand-shaped to a long, straight, and smooth-surface dough piece. Fermentation and final proofing were separately performed for 90 min and 45 min in a fermentation cabinet (SINMAG, Wuxi, China) at a temperature of 30 ℃ and relative humidity of 85%. During fermentation, the dough was punched to squeeze out gas at 55 min and 80 min after onset of fermentation. After fermentation, the dough was molded and placed in baking tins (size of 12.5 × 6.9 × 5.8). The loaf was then baked in a deck oven (SINMAG, Wuxi, China) for 20 min at 215 ℃. The loaves were cooled down and removed from the tins at room temperature. Finally, the loaves were stored in plastic bags at room temperature for further analysis.

After the bread cooled, the specific volume (ml g^−1^) of the bread was determined as: bread volume/bread weight (AACC, 2000) [18]. The crumb texture of bread (hardness, cohesiveness, chewiness, springiness, resilience) was determined by a TA-XT plus texture analyzer (Stable Micro Systems, Surrey, UK) using a P/50 probe. In brief, bread was sliced horizontally, and a piece of 25 mm height bread was compressed to 50% original height. Each sample was run at a 2.0 mm s^−1^ pre-test speed and a 1.0 mm s^−1^ post-test speed with a force of 5 g and a waiting time of 5 s between the first and second compression.

Sensory analysis was performed by a panel of ten trained judges from the laboratory. Bread was presented in sealed pouches coded with different numbers to panelists and scored according to the method of GB/T 14611-2008 (2008) [27].

### 2.9. Data Analysis

All data were subjected to the one-way ANOVA using the SPSS 10.0 software package (SPSS Chicago, IL, USA). ANOVA mean comparisons were performed in terms of the least significant difference (LSD), at the significance level of *p* < 0.05.

## 3. Results

### 3.1. Yield and Its Components

Sulfur fertilizer significantly increased the spike number and grain yield of wheat (Table 1). When compared to being applied as basal fertilization, the application of S and Cys at the jointing stage had more noticeable impacts on spike number and yield. In comparison with S_0_, the spike number and yield of Cys(J) increased by 13.3% and 7.9%, respectively. Similarly, S(J) increased the spike number and yield above S_0_ by 12.5% and 7.8%, respectively. The application of S as basal fertilization also showed a positive effect on yield, which was attributed to the higher spike number.

### 3.2. Contents of Protein Components

The contents of albumin, globulin, gliadin, and glutenin were quantified, respectively. Overall, sulfur and cysteine application had different effects on protein components (Figure 1). Both sulfur and cysteine application increased albumin, gliadin, and glutenin significantly in comparison with the control. The effect of base application was better than that of application at jointing stage on albumin and glutenin contents. Namely, in comparison with S_0_, S(B) and Cys(B) increased albumin by 47.9% and 50.2%, respectively. Similarly, gliadin content was increased by 35.5% and 39.7% in S(B) and Cys(B), respectively. By contrast, the effect of fertilization application at jointing stage was better than that of basal fertilizer application. Cys(J) had the greatest impact on glutenin content, increasing it by 24.4%. In terms of fertilization type, the application of cysteine was better than inorganic sulfur on the improvement of each component.

### 3.3. Contents of HMW-GS and LMW-GS

Both HMW-GS and LMW-GS were increased by cysteine or sulfur fertilizer (Figure 2). Basal fertilizer performed better in increasing HMW-GS content than the jointing stage application did. S(B) and Cys(B) increased the HMW-GS content by 29.7% and 34.7%, respectively. In contrast, fertilizer treatment at the jointing stage had a greater effect on boosting LMW-GS concentration. S(J) and Cys(J) increased the LMW-GS content by 21.7% and 22.7%, respectively. In addition, cysteine fertilizer showed better effects in increasing the contents of HMW-GS compared with sulfur fertilizer in two stages, but the increasing effect on the contents of LMW-GS was inconsistent.

### 3.4. Content of GMP and GMP Particle Size Distribution

Therefore, the changing trend of GMP content is in line with LMW-GS content. Both sulfur and cysteine fertilizer improved GMP content in grains (Figure 3). When comparing fertilization treatments at different stages, the jointing stage treatment improved GMP content more than the basal fertilizer treatment. Cys(J) had the best effect among different sulfur fertilizer treatments, increasing GMP content by 43.5%.

GMP particle size distribution presented two obvious large peaks and two obvious small peaks (Figure 4). The application of sulfur and cysteine fertilizer improved the average particle size of GMP particle volume distribution (Table 2), but only sulfur treatments produced statistically significant improvements. S(J) showed the best improvement effect on the average particle size, with an increase of 71.5%. In comparison with S_0_, the proportion of particles < 10 μm decreased significantly by 28.5% under S(J) treatment, and the proportion of 10–100 μm and >100 μm particles increased by 17.2% and 44.6%, respectively.

### 3.5. Sulfur Content and Nitrogen/Sulfur (N/S) Ratio

The sulfur and cysteine application significantly increased grain nitrogen and sulfur content (Figure 5). The effects of fertilization application at jointing stage were better than those of application as basal fertilization. Among different types of fertilizer treatments, Cys(J) treatment had the largest effect on increasing the content of nitrogen and sulfur in grains, which increased 26.3% and 36.4%, respectively. The variation trend of N/S ratio reflected the changes in nitrogen and sulfur content in grains to a certain extent. Among fertilization treatments at different stages, the effect of fertilizer application at the jointing stage was better than that of basal application. Among different sulfur fertilizer treatments, Cys(J) treatment had the most significant effect, and the N/S ratio decreased by 7.5%.

### 3.6. Gluten Quality

The application of sulfur fertilizer increased the content of wet gluten and dry gluten and sedimentation value (Table 3). Overall, the modification effects on gluten content and sedimentation value were more significant of fertilizer application at jointing stage than those of application as basal fertilization. Among different treatments, Cys(J) showed the best regulation effects, rising wet gluten, dry gluten, and sedimentation value by 38.6%, 10.9%, and 60.5%, respectively.

### 3.7. Bread Baking Quality

The application of sulfur fertilizer had a significant effect on the bread quality parameters (Table 4). The application of sulfur fertilizer significantly reduced the hardness of bread, and the degree of the reduction of the hardness of bread was greater than that of the base sulfur treatment. Among the different types of sulfur fertilizer, Cys(J) had the greatest effect, and the reduction of hardness was up to 69.3%. The effect of sulfur fertilizer on the chewiness of bread was basically the same as that of bread hardness. The application of sulfur fertilizer significantly reduced chewiness, and the effect of top fertilizer on chewiness was better than that of base fertilizer. Among different types of sulfur fertilizer, the effect of Cys(J) on chewiness was the greatest, and the reduction range of chewiness was 69.1%. The effect of sulfur fertilizer on the cohesion, elasticity, and resilience of bread was relatively small, but it also improved the elasticity of bread to some extent. Although sulfur application reduced the cohesion and resilience of some treatments, there was no significant difference among treatments.

The improvement effect of top-dressing on specific volume was better than that of base fertilizer in different periods of sulfur application. Cys(J) showed the greatest effect among different types of sulfur fertilizer treatments, and the volume increased by 109.8% compared with the control.

Ten trained judges were asked to score the bread sensory evaluation (Table 5). The results showed that sulfur fertilizer had a certain effect on the sensory score of bread, that the improvement effect of top sulfur fertilizer was better than that of base sulfur fertilizer, and that the improvement effect of cysteine was the same as that of inorganic sulfur fertilizer, which increased the sensory score of bread by 10%. The analysis of elastic flexibility showed that the effect of topdressing sulfur was better than that of basal sulfur, that the effect of cysteine was the same as that of inorganic sulfur, and that the increase of elastic flexibility reached 17.6%. There was no significant difference in the influence of other sensory evaluation indexes, but Cys(J) had the highest overall score.

## 4. Discussion

### 4.1. Effects of Cysteine and Inorganic Sulfur Application at Different Stages on Protein Quality of Wheat

Sulfur, as well as nitrogen, phosphorus, and potassium, can improve the quality of wheat, which is one of the key nutrient elements that regulates plant growth and grain filling, and acts as a substrate for protein synthesis, especially altering grain protein content [16]. The results of this experiment showed that sulfur application increased the content of protein components in wheat grains. The improvement effect of base sulfur application on albumin and gliadin was better than that of top sulfur application, and the improvement effect of top sulfur application on globulin and gluten was better than that of base sulfur application. Bonnot et al. [28] believed that the combination of nitrogen fertilizer and sulfur fertilizer can improve the protein content and change the protein composition ratio to a certain extent, which is similar to the results of this study. Zhao et al. [29] showed that application of sulfur could significantly increase grain protein content and increase grain yield. The protein and gluten contents of wheat were significantly increased by applying 60 kg ha^−1^ cysteine sulfur at the jointing stage. Different wheat varieties showed different increases, which may be due to the differences in the type of wheat gluten selected and the differences of planting environments.

Glutenin macropolymer (GMP) is a key factor affecting the rheological properties and baking quality of dough. It is composed of HMW-GS and LMW-GS bonded and polymerized by disulfide bond [30]. The results of this study showed that sulfur application increased the content of HMW-GS, LMW-GS, and GMP. The improvement effect of top application of sulfur fertilizer on LMW-GS and GMP was better than that of base application of sulfur fertilizer at the elongation stage, while the improvement effect of base application of sulfur fertilizer on HMW-GS was better than that of top application of sulfur fertilizer, which may be because the base application of sulfur fertilizer was used for sulfur metabolism of the whole plant due to its early application period. At the same time, there was a certain amount of fertilizer loss, and less sulfur was used for grain metabolism at grain filling stage. However, sulfur fertilizer was applied late at the jointing stage, which was more used for sulfur metabolism of grains [17,31,32]. LMW-GS is a sulfur-rich protein, which is the main component of GMP [33,34], so the improvement of LMW-GS and GMP in the jointing stage is obvious. At the jointing and booting stage, the sulfur uptake peak of wheat, and the soil sulfur supply capacity of basal sulfur fertilizer treatment was less than that of topdressing treatment [17]. Wieser et al. [35] showed that sulfur deficiency resulted in an increase in the proportion of HMW-GS and a decrease in the proportion of LMW-GS, so basal sulfur application increased HMW-GS content.

A few studies reported that the diameters of GMP particles were influenced by genotypes and environment, and the diameters of GMP particles were in the range of 1–300 µm [20,36]. The volume percentage of GMP particles < 60 µm decreased within sulfur rates from 30–90 kg ha^−1^ under lower N treatments. The volume percentage of GMP particles > 60 μm increased within the sulfur rates from 30–60 kg ha^−1^, while decreased when excessive sulfur rate of 90 kg ha^−1^ was applied [34]. The results of this study showed the sulfur application significantly increased the proportion of GMP with large particle size (>100 μm, 10–100μm), increased the average particle size of GMP, and decreased the proportion of small particle size (<10 μm). It is suggested that appropriate sulfur fertilizer was favorable for the formation of large GMP particles. The effect of sulfur application on grain sulfur content was the same as that of GMP, which was consistent with the results of previous studies [5,33].

### 4.2. Effects of Cysteine and Inorganic Sulfur Application at Different Stages on Wheat Processing Quality

The quality of wheat grain protein determines the baking quality. The results showed that sulfur application could increase the dry and wet gluten content and sedimentation volume of flour. The result of Wilson et al. [37] showed that under high nitrogen conditions, sulfur application significantly increased wet gluten content and sedimentation volume, thus affecting the baking quality of bread, which was consistent with the conclusion of this experiment. Hardness of bread refers to the force needed to obtain the specified deformation of bread samples; chewiness refers to energy required to chew bread samples into a stable state when swallowing [38]. Hardness and chewiness generally have a negative correlation with bread baking quality [39]. Cohesiveness reflects the internal cohesion of the sample and the ability to resist external damage, resilience reflects the recovery degree of the deformed sample under the same speed and pressure conditions, springiness refers to the ratio of deformation samples to the height before deformation after removal of pressure [40]. Cohesiveness, resilience, and springiness are generally positively correlated with bread quality [41]. At the same time, sulfur application significantly reduced the hardness and chewiness of bread and increased the volume of bread. The effect of sulfur application at jointing stage was better than that of base sulfur application, and the effect of cysteine at the same time was better than that of inorganic sulfur application, which showed that the application of sulfur fertilizer significantly improved the texture characteristics of bread.

The results of Tao et al. [42] showed that the volume of bread under sulfur treatment increased significantly, and sulfur application could improve the baking quality of wheat. De Ruiter and Martin [43] reported that under the condition of 46 kg ha^−^^1^ of sulfur application, the volume of bread increased by more than 6%. The study of Unbehend et al. [44] showed that grain gluten content was positively correlated with bread volume. The studies of other authors [45,46,47] showed that the increase of glutenin content was beneficial to the baking quality of bread. Ortolan et al. [48] reported that protein quality is strongly correlated with bread sensory score, and better protein quality determines higher bread score. In this experiment, the application of cysteine at the jointing stage showed the best effect on improving the baking quality of wheat, which may be because of the high efficiency of sulfur absorption and utilization of wheat and the better accumulation of glutenin as sulfur-rich protein. The baking volume of bread is directly related to gluten quality, and the gluten quality of wheat grain under cysteine treatment at jointing stage is the best, giving the maximum volume of bread.

## 5. Conclusions

The effect of sulfur application on grain protein quality and the processing quality of wheat was significant. The effects of sulfur application on protein quality showed that sulfur application at the jointing stage had significant effects on grain yield, protein composition and content, LMW-GS, GMP content, and grain size distribution of wheat. Among different fertilizer treatments, the contents of globulin, glutenin, LMW-GS, and GMP were significantly increased by topdressing cysteine, and the percentage and average particle size of the large particle size distribution of GMP particles were increased. The effect of sulfur application on processing quality showed that sulfur application at different stages significantly increased the dry and wet gluten content and SDS sedimentation value of flour, and improved the hardness, chewiness, volume, and sensory evaluation score of bread. Among different fertilizer treatments, the application of cysteine significantly increased the dry and wet gluten content and SDS sedimentation value. It significantly improved the baking quality.

## Figures and Tables

**Figure 1 foods-11-03252-f001:**
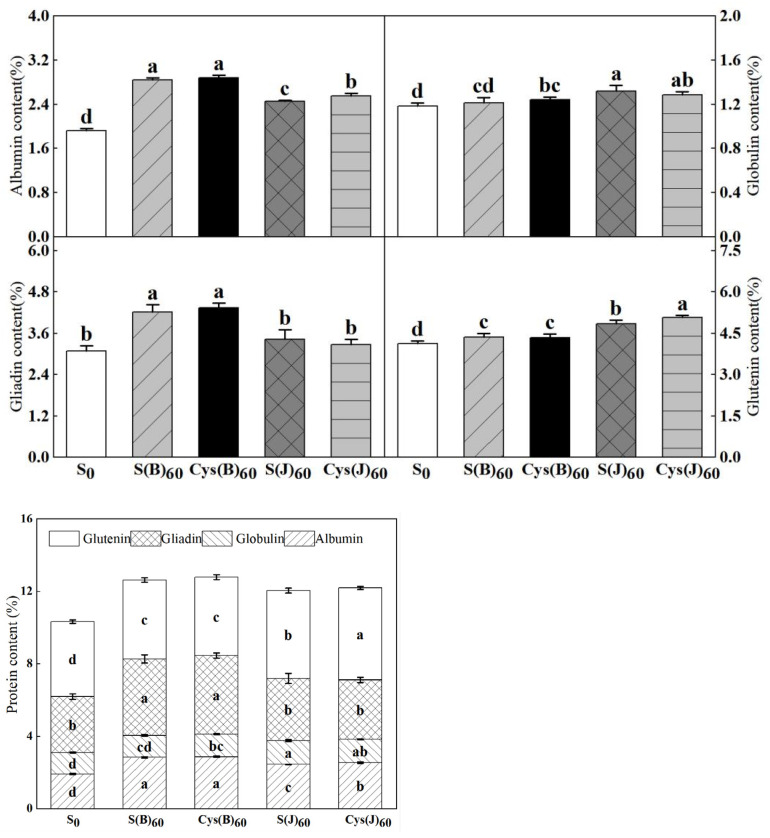
Effects of cysteine and inorganic sulfur at different stages on contents of protein and its components in flour. Different small letters in the same column of each treatment are significantly different at a 0.05 probability level.

**Figure 2 foods-11-03252-f002:**
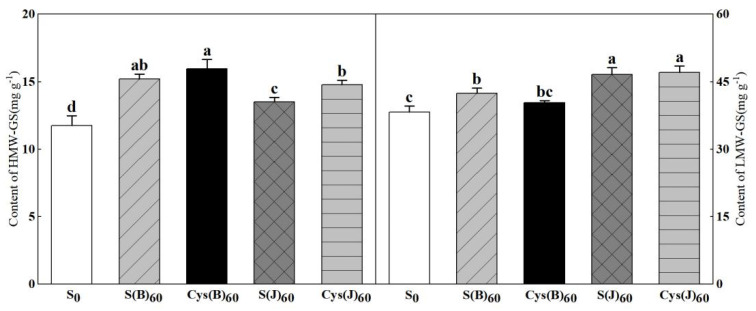
Effects of cysteine and inorganic sulfur at different stages on contents of total HMW-GS and LMW-GS in flour. Different small letters in the same column of each treatment are significantly different at a 0.05 probability level.

**Figure 3 foods-11-03252-f003:**
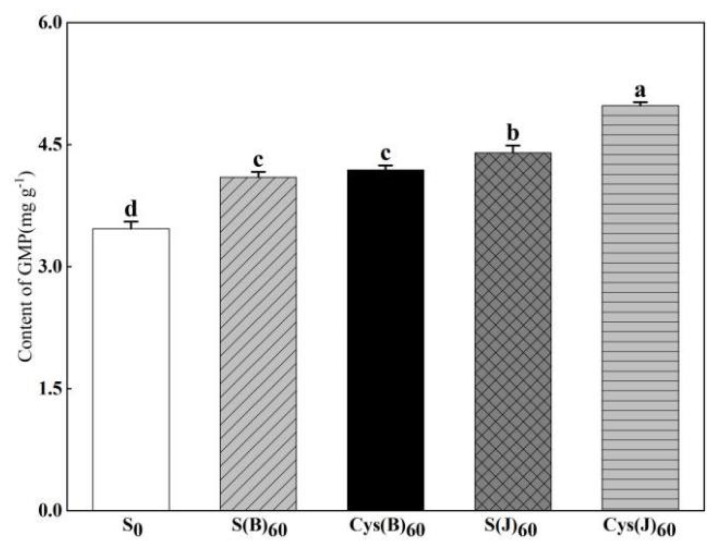
Effects of cysteine and inorganic sulfur at different stages on the contents of GMP in flour. Different small letters in the same column of each treatment are significantly different at a 0.05 probability level.

**Figure 4 foods-11-03252-f004:**
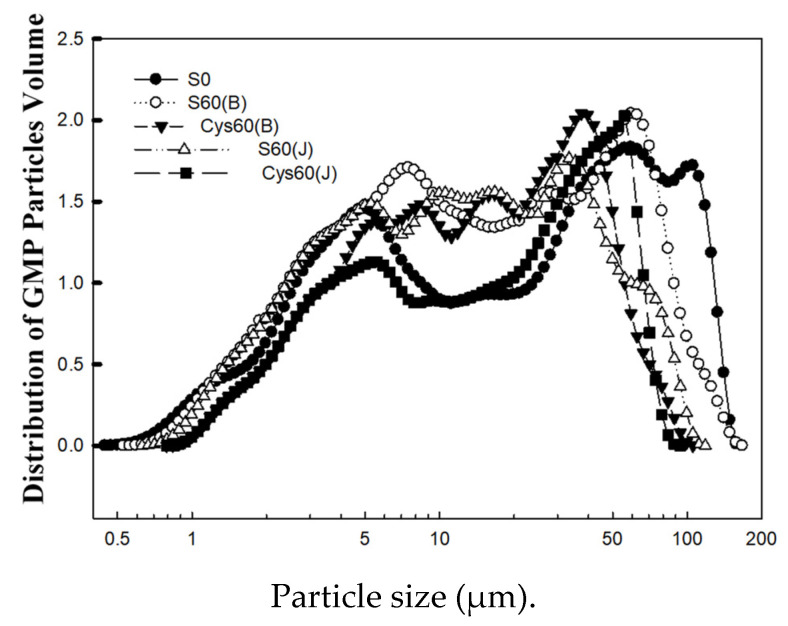
Effects of cysteine and inorganic sulfur at different stages on GMP particles volume in flour. Different small letters in the same column of each treatment are significantly different at a 0.05 probability level.

**Figure 5 foods-11-03252-f005:**
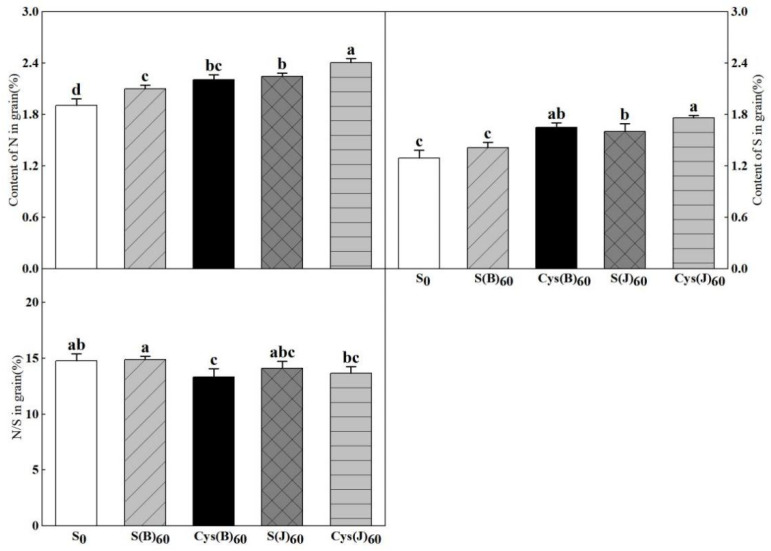
Effects of cysteine and inorganic sulfur at different stages on contents of nitrogen and sulfur content and the ratios of nitrogen to sulfur ratio in flour. Different small letters in the same column of each treatment are significantly different at a 0.05 probability level.

**Table 1 foods-11-03252-t001:** Effects of cysteine and inorganic sulfur application at different stages on wheat yield components.

Treatment	Spike Numbers 1 × 10^4^ ha^−1^	Kernels per Spike	1000-Kernels Weight (g)	Yield (kg ha^−1^)
S_0_	375 ^c^	53.80 ^a^	44.64 ^b^	7361 ^b^
S(B)_60_	421 ^a^	48.58 ^b^	45.33 ^ab^	7811 ^a^
Cys(B)_60_	392 ^b^	49.85 ^b^	44.15 ^b^	7304 ^b^
S(J)_60_	422 ^a^	50.35 ^ab^	44.68 ^b^	7942 ^a^
Cys(J)_60_	425 ^a^	51.57 ^ab^	46.13 ^a^	7944 ^a^

Note: Different lowercase letters in column indicate a significant difference between different treatments in the same column at *p* < 0.05 level.

**Table 2 foods-11-03252-t002:** Effects of cysteine and inorganic sulfur application at different stages on distribution of GMP particles volume.

Treatment	<10		10–100		>100		Mean Diameter
	%	μm	%	μm	%	μm	
S_0_	43.27 ^b^	4.61 ^ab^	53.28 ^b^	39.17 ^ab^	1.13 ^c^	108.3 ^b^	23.12 ^c^
S(B)_60_	42.36 ^c^	4.55 ^b^	52.84 ^b^	40.55 ^b^	4.80 ^ab^	117.1 ^ab^	29.06 ^b^
Cys(B)_60_	43.25 ^b^	4.66 ^ab^	54.07 ^b^	37.14 ^b^	2.70 ^b^	118.0 ^ab^	25.27 ^c^
S(J)_60_	30.94 ^d^	4.71 ^a^	62.44 ^a^	48.46 ^a^	6.17 ^a^	119.4 ^a^	39.66 ^a^
Cys(J)_60_	44.25 ^a^	4.67 ^ab^	53.43 ^b^	36.65 ^b^	2.24 ^b^	115.3 ^ab^	24.27 ^c^

Note: Different lowercase letters in column indicate a significant difference between different treatments in the same column at *p* < 0.05 level.

**Table 3 foods-11-03252-t003:** Effects of cysteine and inorganic sulfur application at different stages on gluten indices and sedimentation volume of flour.

Treatment	Wet Gluten (%)	Dry Gluten (%)	Gluten Index	Sedimentation Volume(mL)
S_0_	33.46 ^c^	13.65 ^c^	79.7 ^a^	43.20 ^c^
S(B)_60_	39.21 ^bc^	14.58 ^b^	72.2 ^b^	62.30 ^b^
Cys(B)_60_	39.58 ^bc^	14.43 ^b^	70.7 ^b^	64.60 ^b^
S(J)_60_	43.09 ^b^	15.23 ^a^	74.0 ^ab^	67.27 ^ab^
Cys(J)_60_	46.47 ^a^	15.25 ^a^	74.1 ^ab^	69.33 ^a^

Note: Different lowercase letters in column indicate a significant difference between different treatments in the same column at *p* < 0.05 level.

**Table 4 foods-11-03252-t004:** Effects of cysteine and inorganic sulfur application at different stages on texture parameters of bread.

Treatment	Hardness (g)	Chewiness (N)	Cohesiveness	Springiness	Resilience	Specific Volume (cm ^3^ g ^−1^)
S_0_	3476 ^a^	2167 ^a^	0.70 ^a^	0.89 ^a^	0.31 ^a^	1.23 ^c^
S(B)_60_	1862 ^b^	1130 ^b^	0.68 ^a^	0.90 ^a^	0.29 ^a^	2.08 ^b^
Cys(B)_60_	1402 ^c^	878 ^c^	0.68 ^a^	0.92 ^a^	0.29 ^a^	2.31 ^ab^
S(J)_60_	1421 ^c^	876 ^c^	0.67 ^a^	0.92 ^a^	0.29 ^a^	2.45 ^a^
Cys(J)_60_	1054 ^d^	669 ^d^	0.68 ^a^	0.91 ^a^	0.29 ^a^	2.58 ^a^

Note: Different lowercase letters in column indicate a significant difference between different treatments in the same column at *p* < 0.05 level.

**Table 5 foods-11-03252-t005:** Effects of cysteine and inorganic sulfur at different stages on sensory evaluation of bread.

Treatment	Appearance Color	Surface Texture	Inside Color	Smoothness	Structure	Flexibility
S_0_	4.0 ^ab^	4.0 ^b^	4.6 ^a^	8.0 ^b^	22.3 ^ab^	8.5 ^b^
S(B)_60_	3.8 ^b^	4.0 ^b^	4.6 ^a^	8.0 ^b^	22.8 ^ab^	8.0 ^c^
Cys(B)_60_	4.0 ^ab^	4.3 ^a^	4.5 ^ab^	8.3 ^a^	23.3 ^a^	9.2 ^ab^
S(J)_60_	4.4 ^a^	4.2 ^ab^	4.5 ^ab^	7.6 ^c^	22.0 ^b^	10.0 ^a^
Cys(J)_60_	4.4 ^a^	4.0 ^b^	4.3 ^b^	8.4 ^a^	23.0 ^a^	10.0 ^a^

Note: Different lowercase letters in column indicate a significant difference between different treatments in the same column at *p* < 0.05 level.

## Data Availability

The date are available from the corresponding author.

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
