# Peer review of "Effects of Cysteine and Inorganic Sulfur Applications at Different Growth Stages on Grain Protein and End-Use Quality in Wheat"

_foods, 2022, doi:10.3390/foods11203252_

Round 1

Reviewer 1 Report

Throughout the paper you refer to jointing stage, but I double checked literature, there is not such a stage. I am not sure wat you referring to but this must be corrected. Then you use a unit kg hm-2, but this is not a standard unit, so it is not clear what it is. You have to explain this. 

In line 84 you say "the basic seedlings of mu" but what is mu? 

The research is completed so reporting must be done in the past tense (grains were harvested, impurities were removed etc.).

2.2. Contents of protein and protein components: your methodology must be described in such a way that other researchers can repeat the experiment. There is far too little detail, and no reference of the methods you used. 

A sentence cannot start with a number.

Line 135: What is a semi-trace Kjeldahl method?

2.7. Bread baking quality: this was totally inadequately described. You have to give a reference and at least describe all the characteristics you measured. In the results you reported on a number of measured characteristics which you did not even mention here. 

Data analysis: why would you do a t test when you do an ANOVA? The ANOVA will provide a LSD for comparison of values. 

If the materials and methods are not clearly described with sufficient detail, it is difficult to follow the rest of the paper, and it also puts into question your results. I have annotated the pdf as far as possible.  

Author Response

Dear,

Thank you very much for giving us an opportunity to revise our manuscript, we also appreciate your valuable comments on our manuscript. The main corrections in the paper and the responses to your comments are as following:

Point 1: Throughout the paper you refer to jointing stage, but I double checked literature, there is not such a stage. I am not sure wat you referring to but this must be corrected. Then you use a unit kg hm-2, but this is not a standard unit, so it is not clear what it is. You have to explain this.  

 Response 1:Thanks for your advice, firstly, when the temperature rises above 10℃ in spring, the interval between the base of wheat begins to elongate, and when the joints are exposed to the ground 1. 5-2. 0 cm, it is called wheat jointing stage. Secondly, I am sorry that I did not consider the use of international units. A unit kg hm-2 means kg ha1. Finally, I have replaced the units that appeared in the article with international units.

Point 2: In line 84 you say "the basic seedlings of mu" but what is mu?

Response 2: Thanks for your advice, I am sorry that the unit did not express clearly. One hectare equals fifteen acres. What I want to say here is that I set the seeding density to 240×104 ha− 1. I have replaced the units that appeared in the article with international units.

Point 3: The research is completed so reporting must be done in the past tense (grains were harvested, impurities were removed etc. ).

Response 3: I am sorry for such a low grammatical error, the tense problems in the article have been revised in the manuscript.  

Point 4: 2. 2. Contents of protein and protein components: your methodology must be described in such a way that other researchers can repeat the experiment. There is far too little detail, and no reference of the methods you used.  

 Response 4: Thanks for your advice, detailed steps have been added to the manuscript at your suggestion.

Specific steps are as follows : according to the solubility of proteins in various solvents, four protein components, namely, albumin, globulin, gliadin and glutenin, were extracted in sequence according to a previous study. Extraction of albumin was performed as follows:One gram of lour was weighed and placed in a test tube, and 10 ml of distilled water was added. Then, the mixture was oscillated for 30 min. The supernatant was centrifuged at 4000 r / min for 5 min and transferred to a sterilizing tube. Repeat the operation for 4 times and steam dry. The extracts of globulin, gliadin, and glutelin were obtained in 10% NaCl, 70% (w/v) ethanol , and 0. 2% NaOH,  respectively. The grain N content was determined using the semi-micro Kjeldahl method, which was multiplied by the coefficient of 5. 7 to get the content of protein and protein components.

Point 5: A sentence cannot start with a number.

 Response 5: I am sorry for such a low grammatical error, errors of the same type in the article have all been corrected in the manuscript.

Point 6: Line 135: What is a semi-trace Kjeldahl method?

 Response 6: Thanks for your advice, the semi-micro Kjeldahl method is one of the conventional methods for determining total nitrogen in geochemical sample. The principle of this method is as follows, the sample material is digested by sulfuric acid, transferred to Kay tube, alkalized with sodium hydroxide solution, heated distillation to escape ammonia, absorbed by boric acid solution, titrated with hydrochloric acid standard solution, and calculated the ammonia content in the sample. 

Point 7: 2.7.Bread baking quality: this was totally inadequately described. You have to give a reference and at least describe all the characteristics you measured. In the results you reported on a number of measured characteristics which you did not even mention here.  

 Response 7: Thanks for your advice, I put together a summary of how the bread is made and how it is measured. The following steps have been added to the manuscript.

Breads were prepared using the straight-dough method according to Chinese National Standard GB/T 14611-2008 (2008). The formula contains 100 g flour, 6 g sugar, 4 g skim milk powder, 3 g shortening, 1.8 g instant dry yeast, 1.5 g salt, 0.2 g wheat malt flour and 4 mg ascorbic acid. Salt and sugar were dispersed in water. The mixture solution of salt and sugar was added to the premixed dry ingredients. Cohesive dough was prepared by hand. Amount of water addition, mixing and proof time were adjusted according to the performance of the dough. The dough was hand-shaped to a round, long and straight, and smooth-surface dough piece.Fermentation and final proofing were separately performed for 90 min and 45 min in a fermentation cabinet (SINMAG, Wuxi, China) controlling at a temperature of 30℃and relative humidity of 85%. During fermentation, the dough was punched to squeeze out of gas at 55 min and 80 min after onset of fermentation. After fermentation, the dough was molded and placed in baking pans(size of 12.5×6.9×5.8). The loaf was then baked in a deck oven (SINMAG, Wuxi, China) for 20 min at 215 ℃.The loaves were cooled down and removed from the tins and cooled down at room temperature. Finally the loaves were stored in plastic bags at room temperature for bread quality analysis.

After the bread was baked and cooled, the specific volume (ml/g) of the bread was determined as: bread volume/bread weight (AACC, 2000). The crumb texture of bread (hardness, cohesiveness, chewiness, springiness, resilience) was determined by a TA-XT plus texture analyzer (Stable Micro Systems, Surrey, UK) using a P/50 probe. In brief, bread was sliced horizontally, and a piece of 25 mm height bread was compressed to 50% original height. Each sample was run at a 2.0 mm/s pre-test speed and a 1.0 mm/s post-test speed with a force of 5 g and waiting time of 5 s between the first and second compression.

Sensory analysis was performed by a panel of ten trained judges from the laboratory.

Point 8: Data analysis: why would you do a t test when you do an ANOVA? The ANOVA will provide a LSD for comparison of values.  

 Response 8: Thanks for your advice, I update the data processing method as follows, All data were subjected to one-way ANOVA using the SPSS Version 10.0. ANOVA mean comparisons were performed in terms of the least significant difference (LSD), at the significance level of P < 0.05.

Point 9: If the materials and methods are not clearly described with sufficient detail, it is difficult to follow the rest of the paper, and it also puts into question your results. I have annotated the pdf as far as possible.   

 Response 9: Thanks for your advice, your advice is very helpful to me, according to your advice, the material and methods section has been described as much as possible in detail, thank you again. 

Other changes in the manuscript were also marked in red color.

We tried our best to improve the manuscript and hope that the correction will meet with approval.

Once again, thank you very much for your comments and suggestions.

Yours sincerely,

Jian Cai

Reviewer 2 Report

Comments and Suggestions for Authors

General appreciation

Short article presenting the effect of sulfur addition, in two forms and at two different stages of wheat development, on the protein composition of soft wheat, and the impact of this composition on the baking quality of the flour. The article contains few references in spite of an abundant literature on the subject, and the discussion is limited. Moreover, it does not present any major advance, it only confirms the existing data on the effect of sulfur fertilization.

The materials and methods are not precise enough to repeat experiments.  The agronomic description of the experimental design is not very precise and has probably not been validated by an agronomist. The same is true for section 2.2, which was not written by a cereal chemist. The feasibility of determining the nitrogen content of flour pellet by Biuret reagent is questionable. No indication is given on the method used to separate HMW-GS and LMW-GS, one can assume that it is SDS-PAGE. Similarly, there is a lack of information in the legends, figures and tables (e.g., in Table 3). Furthermore, no easily verifiable information is available on the technological tests (milling, flour composition, characteristics of the bread-making test...). Too much information is missing or not detailed enough to fully understand the experiments and results.

The results are well described but there is a lack of statistical information in the figures to indicate whether the variations are significant or not (e.g. letters at the top of each histogram bar). There is also no information on the number of analyses performed to obtain an analytical result.

The discussion is weak compared to the existing literature.

Abstract

Line 11: "combined" is not an appropriate term

Line 23: please explain/justify "damage degree"

Introduction

Line 43: ref [5] is interesting but it not sufficient to support " The number of free sulfhydryl groups determines the formation of high molecular weight glutenin subunit (HMW-GS), low molecular weight glutenin subunit (LMW-GS) and glutenin macropolymer". Please provide more pertinent references.

Line 47: ref [7] and [8] do not provide information on flour quality. Please find new references.

Line 65: please define " The contents of protein components".

Materials and methods

Line 73: please explain what is a medium gluten variety.

Line 80: please express fertilizer values per ha. The same for yield and spike number.

Line 81: what is "jointing application"? What is "jointing"?

Line 83: what was the experimental design used?

Line 84: what do you mean by "After sowing, the basic seedlings of mu were 240×104 hm-2."?

Line 88: "in the mature period", not the right term

Table 1: not necessary. The different conditions are already described in the text.

Line 99: " Extraction of protein components was conducted according to modern plant physiological test guidelines". Please insert references or international standards.

Line 100: please add details to explain " The continuous extraction method was used to extract albumin, globulin, gliadin and glutenin in grains with distilled water, 10% NaCl, 70% ethanol and 0.2% NaOH solution in turn".

Line 103: ref [13] do not provide information for " by semi-trace Ketamine method".

Line 106: It is not clear for me on what part you measure protein content. In ref [14] Don et al. considered that GMP is the middle layer. Please could you explain how do you succeed to take away this layer when performing a micro-assay? How do you express the result of the Biuret assay?

Line 120-133: please provide additional details on the quantification of the different proteins groups and subcategories (albumin, globulin, gliadin, HMW-GS and LMW-GS).

Line 137: bread making and bread quality evaluation are insufficiently described. No reference to international standards recognized in cereal sciences (ISO, ICC, ACCI) or literature. There is no description of bread sample taken to texture analysis (diameter, thickness). There is no indication on the panel (repartition man/woman, age) and on the test (type, conditions, sample, statistical analysis). Please add more details.

Methods such SDS sedimentation volume measurement, gluten preparation and GI assay are not described. Please provide details on methods, equipment or references.

Results

Table 2: please reduce the number of decimal places. Two decimal places are not necessary for this type of data. The same for all almost of the others tables.

Line 150: please detail CK?

Fig 1 and fig 2: please indicate the results of the statistical analysis on the graph? The expression of the result is not clear for me. Please indicate how do you express the data? Are they % of the total protein content?

Fig 1: proteins are explained in % but in fig 2, HMW-GS and LMW-GS are explained in mg.g-1, why did you make a change?

Line 185: "with LMW-GS as the main part of GMP".  According to Don et a. (2006), Journal of Cereal Science, 44(2), 127-136, HMW-GS are considered to be the main contributor to the GMP.

Table 3: Mu m? Is it the average particle size in µm?

Line 262: Please define the sensory score, how did you calculate it?

Discussion

Line 275: "The improvement effect of base sulfur application on albumin and gliadin was better than that of top sulfur application, and the improvement effect of top sulfur application on globulin and gluten was better than that of base sulfur application". To illustrate that it would be better to change Fig 1 and express the protein content of the different subclasses in proportion of the total protein content.

Line 289: " At the same time, there was a certain 289 amount of fertilizer loss, and less sulfur was used for grain metabolism at grain filling 290 stage.". Please provide some references to support this statement.

Line 302: Dai et al. [23] = Yan et al.  in the Bibliography

Line 325: " The study of Hu et al. [28] showed that the increase of glutenin content was beneficial to the baking quality of bread". Please provide more relevant references to support this statement.

Author Response

Dear,

Thank you very much for giving us an opportunity to revise our manuscript, we also appreciate your valuable comments on our manuscript. The main corrections in the paper and the responses to your comments are as following:

 Point 1: Short article presenting the effect of sulfur addition, in two forms and at two different stages of wheat development, on the protein composition of soft wheat, and the impact of this composition on the baking quality of the flour. The article contains few references in spite of an abundant literature on the subject, and the discussion is limited. Moreover, it does not present any major advance, it only confirms the existing data on the effect of sulfur fertilization.

Response 1: Thanks for your advice, firstly, we do recognize that references are rarely sufficient to support our conclusions, so we have added some relevant references to the original references. This section has been marked in the manuscript. Please check it out. Secondly, the discussion part only considered part at that time, so the modification added the comparison with previous results and the part of bread texture characteristic on the original basis. Finally, about the results, we know that the effect of sulfur fertilizer application at different stages on wheat quality is not clear. Therefore, through this experiment, we studied the regulation of different forms of sulfur fertilizer on wheat glutenin subunits and glutenin-macropolymer, and the effect on bread baking quality, which can provide reference for optimizing wheat cultivation.

Point 2: The materials and methods are not precise enough to repeat experiments. The agronomic description of the experimental design is not very precise and has probably not been validated by an agronomist. The same is true for section 2.2, which was not written by a cereal chemist. The feasibility of determining the nitrogen content of flour pellet by Biuret reagent is questionable. No indication is given on the method used to separate HMW-GS and LMW-GS, one can assume that it is SDS-PAGE. Similarly, there is a lack of information in the legends, figures and tables (e.g., in Table 3). Furthermore, no easily verifiable information is available on the technological tests (milling, flour composition, characteristics of the bread-making test...). Too much information is missing or not detailed enough to fully understand the experiments and results.

Response 2: Thanks for your advice, based on your suggestions, we have expanded the materials and methods section. Firstly, the cultivation process was described in detail in the experimental design. Secondly, with regard to the measurement of indicators, the operation steps were described in detail, including from 2.2 to 2.7. Finally, some parameters of bread texture characteristics were described in detail.

Point 3: The results are well described but there is a lack of statistical information in the figures to indicate whether the variations are significant or not (e.g. letters at the top of each histogram bar). There is also no information on the number of analyses performed to obtain an analytical result.

Response 3: Thanks for your advice, upon receipt of your suggestion, we immediately reintegrate the data to indicate significance, and we plot the results based on three biological replicates per treatment.

Point 4: The discussion is weak compared to the existing literature.

Response 4: Thanks for your advice, according to your suggestion, we have discussed the following points. First of all, in terms of the effect on wheat protein quality, on the basis of the original, increased the comparison with previous results. Secondly, in the aspect of processing quality, the discussion of bread texture characteristics was increased, and the reason of better gluten quality of wheat grain with cysteine sulfur at jointing stage was clarified.

Abstract

Point 5: Line 11: "combined" is not an appropriate term

Response 5: Thanks for your advice, the word "combined" was used at the time because we had a combination of organic sulfur and cysteine sulfur, but it was later found that there was no significant difference between the benefits of the combination treatments and the benefit of the single treatments. In view of the consideration of economic efficiency, we have finally selected these single treatments, and now according to your suggestion we have replaced the word "combined" with "significant." Thank you again for your suggestion.

Point 6: Line 23: please explain/justify "damage degree"

Response 6: Thanks for your advice, sorry for the low level error. This should have been used to describe the effect of sulphur application on dry gluten, wet gluten and sedimentation volume. These are shown in 3.6. This part has been corrected in the manuscript.

Introduction

Point 7: Line 43: ref [5] is interesting but it not sufficient to support " The number of free sulfhydryl groups determines the formation of high molecular weight glutenin subunit (HMW-GS), low molecular weight glutenin subunit (LMW-GS) and glutenin macropolymer". Please provide more pertinent references.

Response 7: Thanks for your advice, on the basis of the original references, we also cited some of the relevant research progress on the effect of sulfur on wheat yield and quality to confirm this part. For example we cited Dai, Z., et al., Transcriptional and metabolic alternations rebalance wheat grain storage protein accumulation under variable nitrogen and sulfur supply [5]. Yu, Z., et al., Impact and mechanism of sulphur-deficiency on modern wheat farming nitrogen-related sustainability and gliadin content [6]. Wu, J., et al., Effects of glu-1 and glu-3 allelic variations on wheat glutenin macropolymer (GMP) content as revealed by size-exclusion high performance liquid chromatography (SE-HPLC) [7] .(line 477-482).

Point 8: Line 47: ref [7] and [8] do not provide information on flour quality. Please find new references.

Response 8: Thanks for your advice, at your suggestion, we have re-searched two articles that are closer to our results, which have been marked in the manuscript. For example we cited Flæte, N.E.S., et al., Combined nitrogen and sulphur fertilisation and its effect on wheat quality and protein composition measured by SE-FPLC and proteomics [9] . GyÅ‘ri, Z., Sulphur Content of Winter Wheat Grain in Long Term Field Experiments [10] .(line 485-488).

Point 9: Line 65: please define " The contents of protein components".

Response 9: Thanks for your advice, according to the solubility of protein in different solutions, Osborne divided protein into four different components: albumin, globulin, gliadin, glutenin. Among them, albumin and globulin are structural proteins, soluble proteins, accounting for about 20% of the total protein, gliadin and glutenin are storage proteins, also known as gluten proteins, which account for about 80% of total protein.

Materials and methods

Point 10: Line 73: please explain what is a medium gluten variety.

Response 10: Thanks for your advice, according to the national wheat validation standard, medium gluten wheat: crude protein content (dry basis) is more than 12.0%, wet gluten content (14% water basis) more than 24.0%, water absorption more than 55%, stability time more than 3.0 minutes, maximum stretch resistance (reference value) over 200, tensile area over 50 cm2

Point 11: Line 80: please express fertilizer values per ha. The same for yield and spike number.

Response 11: Thanks for your advice, I am sorry that the international system of units (SI) was not adopted at that time, and it has now been revised to the text of the unity of the units appear in the international system.

Point 12: Line 81: what is "jointing application"? What is "jointing"?

Response 12: Thanks for your advice, firstly,wheat jointing stage, normal growth of wheat, when the spring temperature rose to more than 10℃, wheat basal internodes began to elongate, internodes exposed to the ground 1.5-2.0cm called jointing. Secondly, in the course of wheat cultivation, basal fertilizer is generally applied before sowing, and topdressing is applied at jointing stage. The concept of confusion appeared in the first manuscript, and has been revised in this revision.

Point 13: Line 83: what was the experimental design used?

Response 13: Thanks for your advice, the field experiments were laid out in a single-factor completely randomized design with three replicates for each treatment, I am very sorry that I did not express myself clearly in the first manuscript, and now I have completed this part.

Point 14: Line 84: what do you mean by "After sowing, the basic seedlings of mu were 240×104 hm-2."?

Response 14: Thanks for your advice, sorry not to have used the international system of units(SI), which has now been revised to “After sowing, the basic seedlings were 240×104 ha−1.”

Point 15: Line 88: "in the mature period", not the right term.

Response 15: Thanks for your advice, "in the mature period" has been changed to “At maturity”.

Point 16: Table 1: not necessary. The different conditions are already described in the text.

Response 16: Thanks for your advice, for the sake of brevity, we have deleted Table 1.

Point 17: Line 99: " Extraction of protein components was conducted according to modern plant physiological test guidelines". Please insert references or international standards.

Response 17: Thanks for your advice, the corresponding references AACC, 2000. Approved Methods of the AACC, 10 ed. St. Paul, MN, USA [18]. have been marked according to your suggestions, which have been revised in the manuscript. (line 504).

Point 18: Line 100: please add details to explain " The continuous extraction method was used to extract albumin, globulin, gliadin and glutenin in grains with distilled water, 10% NaCl, 70% ethanol and 0.2% NaOH solution in turn".

Response 18: Thanks for your advice, extraction of albumin was performed as follows: about 1 g of flour was weighed and placed in a test tube, and 10 ml of distilled water was added. Then, the mixture was oscillated for 30 min. The supernatant was centrifuged at 4000 r / min for 5 min and transferred to a sterilizing tube. Repeat the operation for 4 times and steam dry. The extracts of globulin, gliadin, and glutelin were obtained in 10% NaCl, 70% (w/v) ethanol , and 0. 2% NaOH, respectively. The grain N content was determined using the semi-micro Kjeldahl method, which was multiplied by the coefficient of 5. 7 to get the content of protein and protein components. This section has been added to the manuscript.

Point 19: Line 103: ref [13] do not provide information for " by semi-trace Ketamine method".

Response 19: Thanks for your advice, references have been updated as requested, which includeZhu, J. and K. Khan, Characterization of Glutenin Protein Fractions from Sequential Extraction of Hard Red Spring Wheats of Different Breadmaking Quality [19]. (line 505-506).

Point 20: Line 106: It is not clear for me on what part you measure protein content. In ref [14] Don et al. considered that GMP is the middle layer. Please could you explain how do you succeed to take away this layer when performing a micro-assay? How do you express the result of the Biuret assay?

Response 20: Thanks for your advice,firstly, when Osborne proposed his protein classification system, he pointed out that only part of the glutenin can be extracted by dilute acetic acid, and the other part exists in the residue after extraction and is not soluble in SDS extract. The polymer can only be dissolved after adding a reducing agent to the SDS extract or after sonochemical treatment. This polymer is named "glutenin macropolymer" or simply "GMP." Accordingly, after removing other proteins with 1 ml 1.5% SDS, The color of the purple complex formed by biuret and cupric sulfate in strong alkaline solution is in direct proportion to the concentration of protein in the sample and is independent of molecular weight and amino acid composition of protein.  

Point 21: Line 120-133: please provide additional details on the quantification of the different proteins groups and subcategories (albumin, globulin, gliadin, HMW-GS and LMW-GS).

Response 21: Thanks for your advice, based on your suggestion, we have described the extraction, separation, electrophoresis, fixation, staining, decolorization and quantitative analysis of high and low molecular glutenin subunits in detail in the manuscript. (line 150-160).

Point 22: Line 137: bread making and bread quality evaluation are insufficiently described. No reference to international standards recognized in cereal sciences (ISO, ICC, ACCI) or literature. There is no description of bread sample taken to texture analysis (diameter, thickness). There is no indication on the panel (repartition man/woman, age) and on the test (type, conditions, sample, statistical analysis). Please add more details.

Response 22: Thanks for your advice, according to your suggestion, the bread-making process, bread texture characteristics and bread sensory qualities have been described in detail, (line 173-199). and the relevant literature AACC, 2000 [18]. Approved Methods of the AACC, 10 ed. St. Paul, MN, USA. And Chinese National Standard Management Committee, GB/T 14611, 2008. Inspection of Grain and Oils-Bread-baking Test of Wheat Flour-straight Dough Method. Standards Press of China, Beijing [27]. have been cited to prove it.

Point 23: Methods such SDS sedimentation volume measurement, gluten preparation and GI assay are not described. Please provide details on methods, equipment or references.

Response 23: Thanks for your advice, according to your suggestion, supplement 2.7 in materials and methods, which has been marked in the manuscript. (line 166-174)

Results

Point 24: Table 2: please reduce the number of decimal places. Two decimal places are not necessary for this type of data. The same for all almost of the others tables.

Response 24: Thanks for your advice, according to your suggestion, the question about the number of decimal places has been revised in the manuscript. 

Point 25: Line 150: please detail CK?

Response 25: Thanks for your advice, in the text, CK stands for S0, that is, no sulfur fertilizer is applied throughout the wheat growth stage. To avoid misunderstanding, it has been revised to S0 in the manuscript.

Point 26: Fig 1 and fig 2: please indicate the results of the statistical analysis on the graph? The expression of the result is not clear for me. Please indicate how do you express the data? Are they % of the total protein content?

Response 26: Thanks for your advice, firstly, according to your suggestion, the results of the statistical analysis have been labeled on the graph. Secondly, in figure 1, we used the proportional form to show the proportion of each protein component in the flour protein. In figure 2, the high and low molecular glutenin subunits content represent the amount in the flour tested.

Point 27: Fig 1: proteins are explained in % but in fig 2, HMW-GS and LMW-GS are explained in mg.g-1, why did you make a change?

Response 27: Thanks for your advice, first, the wheat grain is composed of starch and protein, so the protein content can be expressed as a percentage in the grain. Secondly, regarding the high and low molecular glutenin subunits, if expressed as a percentage, it will cause a misunderstanding of the proportion of protein or the ratio of the whole grain, so it is expressed in mg g-1, and from a mathematical point of view, mg g-1 can be converted into a percentage.

Point 28: Line 185: "with LMW-GS as the main part of GMP".  According to Don et a. (2006), Journal of Cereal Science, 44(2), 127-136, HMW-GS are considered to be the main contributor to the GMP.

Response 28: Thanks for your advice, according to Yan, S., et al., Effects of sulphur fertilizer on glutenin macropolymer content and particle size distribution in wheat grain [34]. GMP is cross-linked by HMW-GS and LMW-GS, and LMW-GS are major components of the glutenin aggregations with about 5-6 times more abundant than HMW-GS. So I concluded “LMW-GS as the main part of GMP”.

Point 29: Table 3: Mu m? Is it the average particle size in µm?

Response 29: Thanks for your advice, sorry for such a low level error, it has been revised to μm.

Point 30: Line 262: Please define the sensory score, how did you calculate it?

Response 30: Thanks for your advice, according to the method of GB/T 14611-2008, the form, color, flavor, texture and texture of the bread were scored on a 1-5 scale.

Discussion

Point 31: Line 275: "The improvement effect of base sulfur application on albumin and gliadin was better than that of top sulfur application, and the improvement effect of top sulfur application on globulin and gluten was better than that of base sulfur application". To illustrate that it would be better to change Fig 1 and express the protein content of the different subclasses in proportion of the total protein content.

Response 31: Thanks for your advice, according to your suggestion, the graph has been revised to the type of the proportion of each part.

Point 32: Line 289: " At the same time, there was a certain 289 amount of fertilizer loss, and less sulfur was used for grain metabolism at grain filling 290 stage.". Please provide some references to support this statement.

Response 32: Thanks for your advice, according to your suggestion, two articles are cited to illustrate this point, Assefa, S., W. Haile, and W. Tena, Effects of phosphorus and sulfur on yield and nutrient uptake of wheat (Triticum aestivum L.) [31] and DuPont, F.M., et al., Differential accumulation of sulfur-rich and sulfur-poor wheat flour proteins is affected by temperature and mineral nutrition during grain development [32].

Point 33: Line 302: Dai et al. [23] = Yan et al.  in the Bibliography.

Response 33: Thanks for your advice, according to your suggestion, the correct literature has been cited.Tea, I., et al., Effect of Foliar Sulfur and Nitrogen Fertilization on Wheat Storage Protein Composition and Dough Mixing Properties [33].

Point 34: Line 325: " The study of Hu et al. [28] showed that the increase of glutenin content was beneficial to the baking quality of bread". Please provide more relevant references to support this statement.

Response 34: Thanks for your advice, two articles have been added to the manuscript to support this view.Hu, X., et al., Combined effects of wheat gluten and carboxymethylcellulose on dough rheological behaviours and gluten network of potato–wheat flour-based bread [45].Goesaert, H., et al., Wheat flour constituents: how they impact bread quality, and how to impact their functionality [46]. Leon, E., et al., Pasting properties of transgenic lines of a commercial bread wheat expressing combinations of HMW glutenin subunit genes [47].

Other changes in the manuscript were also marked in red color.

We tried our best to improve the manuscript and hope that the correction will meet with approval.

Once again, thank you very much for your comments and suggestions.

Yours sincerely,

Jian Cai

Reviewer 3 Report

The main objective of the present study was to investigate the effects of two types of fertilizers applied at different stages on grain protein and end-use quality of wheat.

The topic of the study is within the scope of journal, but it has been detected some issues.

The title of the manuscript is not appropriate because it does not reflect the methodology used. Interactive effects of cysteine and inorganic sulfur applications?- The authors investigated a single effects of cysteine and inorganic sulfur applications.

Lines 25-28: Conclusions on the effect of inorganic sulfur and cysteine on grain protein and flour quality are somewhat confusing. Please improve that part in a clearer way, to be consistent with the obtained results.

Lines 46-49: Sulfur fertilizer is not the only factor responsible for improving wet gluten content, bread volume and dough stability. These quality parameters are influenced by variety and location as well. Please rewrite this.

Line 69: Replace “bread product” with “wheat flour”.

Lines 126-132: Specify which buffer refers to HMV-GS and which buffer to LMV-GS extraction.

Line 137: 2.7. Bread baking quality - Please provide more details about methodology used for baking procedure and evaluation of bread quality. It is not common for methods to be presented in this way, especially in the case of textural and sensory analysis where the conditions of the measurements performed are not given (for TPA: force, height of a slice of bread, etc.)

Lines168-169: This is not entirely true; the effect of the type of fertilization is influenced by the time of application. Are these differences statistically significant?

Figures 1,2,3 and 5: Please indicate the meaning of error bars.

Lines 179-180: Please avoid the general conclusions since these are not support by obtained results (for example, in the case of LMW-GS the effect of cysteine fertilizer at both stages are even lower or on the same level compared to the other one fertilizer).

Lines 236-238: There are no statistical differences between the specific volume values regardless of the type of fertilizer.

Lines 247-248: How could that be the best effect? Is it the reduction of hardness positive effect? It would be useful to change "best effect" into "greatest effect" since we do not know the reference value for bread hardness.

Line 320: This is not true for specific volume.

Line 347: Please rewrite this in accordance to the statistically processed results.

Author Response

Dear,

Thank you very much for giving us an opportunity to revise our manuscript, we also appreciate your valuable comments on our manuscript. The main corrections in the paper and the responses to your comments are as following:

Point 1: The main objective of the present study was to investigate the effects of two types of fertilizers applied at different stages on grain protein and end-use quality of wheat. The topic of the study is within the scope of journal, but it has been detected some issues. The title of the manuscript is not appropriate because it does not reflect the methodology used. Interactive effects of cysteine and inorganic sulfur applications?- The authors investigated a single effects of cysteine and inorganic sulfur applications.

Response 1: Thanks for your advice, according to your suggestion, we will make some changes on the basis of the existing one and change it to Effects of cysteine and inorganic sulfur applications at different stages on grain protein and end-use quality in wheat. We set up treatment in the early stage of the experiment, including two kinds of sulfur fertilizer, but from the test results, single application and combined application of the effect of difference is not significant, taking into account the economic benefits, we put the treatment of combined application removed.

Point 2: Lines 25-28: Conclusions on the effect of inorganic sulfur and cysteine on grain protein and flour quality are somewhat confusing. Please improve that part in a clearer way, to be consistent with the obtained results.

Response 2: Thanks for your advice, based on your suggestions, we have made the following revisions, firstly, the effect of topdressing at jointing stage on grain protein quality and processing quality was greater than that of basal application of sulfur fertilizer. Secondly, from the view of different forms of sulfur fertilizer, the effect of cysteine sulfur topdressing at jointing stage on wheat grain protein quality and bread processing quality was better than inorganic fertilizer.  

Point 3: Lines 46-49: Sulfur fertilizer is not the only factor responsible for improving wet gluten content, bread volume and dough stability. These quality parameters are influenced by variety and location as well. Please rewrite this.

Response 3: Thanks for your advice, based on your suggestions, we have made the following revisions. Firstly, the effect of sulfur fertilizer on processing quality was studied. Secondly, the reasons for the different phenomena of different research results are analyzed. Finally, the manuscript reads as follows: the quality of flour can be modified by sulfur fertilizer. Adding sulfur fertilizer can increase wet gluten content and flour settling value of wheat, prolong dough formation time, decrease dough stability time and tensile resistance. The sulfur fertilizer can not only improve the wet gluten content of wheat, but also improve the volume of bread, specific volume and dough stability. Addition of S increased loaf volume significantly at two sites where grain S concentration was also significantly increased and grain N:S ratio decreased. Application of the extra 50 kg ha1 N increased grain protein concentration but did not increase loaf volume at any of the sites. The effect of sulfur on improving quality varies by species and location. (Line 52-56)

Point 4: Line 69: Replace “bread product” with “wheat flour”.

Response 4: Thanks for your advice, this section has been modified in the manuscript.

Point 5: Lines 126-132: Specify which buffer refers to HMV-GS and which buffer to LMV-GS extraction.

Response 5: Thanks for your advice, firstly, we use SDS-PAGE to determine the content of high and low molecular glutenin subunits, so we can not distinguish the high and high molecular gluten subunits. Secondly, after running according to the strip distinction, and then through Quantity One Analysis scanning results.

Point 6: Line 137: 2.7. Bread baking quality - Please provide more details about methodology used for baking procedure and evaluation of bread quality. It is not common for methods to be presented in this way, especially in the case of textural and sensory analysis where the conditions of the measurements performed are not given (for TPA: force, height of a slice of bread, etc.).

Response 6: Thanks for your advice, according to your suggestion, the bread-making process, bread texture characteristics and bread sensory qualities have been described in detail, and the relevant literature has been cited to prove it. (Line 176-202)

Point 7: Lines168-169: This is not entirely true; the effect of the type of fertilization is influenced by the time of application. Are these differences statistically significant?

Response 7: Thanks for your advice, studies have shown that inorganic sulfur as a base fertilizer is easy to lose with rain. So the question you raised is really worth thinking about, so we went back to the statistical analysis of all the data, and the results have been put in the manuscript

Point 8: Figures 1,2,3 and 5: Please indicate the meaning of error bars.

Response 8: Thanks for your advice, firstly, an error bar is a line segment drawn in the direction of the magnitude of the measurement, with the arithmetic mean of the measured value as the midpoint. Half of the length of the line segment is equal to (standard or extended) uncertainty. It indicates that the measurement falls on the rod with some probability (68% or 95%). Secondly, We realized that there was a lack of statistical analysis in our figures, so we carried out statistical analysis of all the data involved and replotted the figures. The new figures have been included in the manuscript.

Point 9: Lines 179-180: Please avoid the general conclusions since these are not support by obtained results (for example, in the case of LMW-GS the effect of cysteine fertilizer at both stages are even lower or on the same level compared to the other one fertilizer).

Response 9: Thanks for your advice, first, we apologize for the general conclusion in the data analysis, and second, it has been revised to the following conclusion ”In addition, cysteine fertilizer showed better effects in regulating HMW-GS comparing with sulfur fertilizer in two stages, but the regulation effect on LMW-GS was inconsistent.” (Line 254-256)

Point 10: Lines 236-238: There are no statistical differences between the specific volume values regardless of the type of fertilizer.

Response 10: Thanks for your advice, according to your suggestion, we have made the following adjustments. firstly, previous studies have shown that sulfur fertilizer plays a great role in the volume of bread, so we have not removed it in this revision. Secondly, we accept your suggestion that this time the bread volume is not analyzed separately, but the texture characteristics of the whole bread are analyzed. The detailed analysis process is in the manuscript.  (Line 325-328)

Point 11: Lines 247-248: How could that be the best effect? Is it the reduction of hardness positive effect? It would be useful to change "best effect" into "greatest effect" since we do not know the reference value for bread hardness.

Response 11: Thanks for your advice, firstly, I am very sorry for this error. We did want to express that the effect of cysteine sulfur is better than other inorganic fertilizers and that jointing fertilizer is superior to base fertilizer, but there is a misnomer. In addition, after careful consideration, we think that the greatest effect you say can express the meaning more accurately, so we have revised it in the manuscript.

Point 12: Line 320: This is not true for specific volume.

Response 12: Thanks for your advice, firstly, this part of the data is reflected in Table 4. Secondly, perhaps because of my unclear expression caused a misunderstanding, in the article "and the effect of cysteine was better than that of inorganic sulfur application" expression means the comparison between different forms of sulfur fertilizer in the same period. Finally, for clarity, the sentence has been changed to “The effect of sulfur application at jointing stage was better than that of base sulfur application, and the effect of cysteine at the same time was better than that of inorganic sulfur application.” (Line 419-421)

Point 13: Line 347: Please rewrite this in accordance to the statistically processed results.

Response 13: Thanks for your advice, firstly, I apologize for the general conclusions that are not supported by the data. Secondly, the language has been reorganized according to the data results. The revised results are as follows “It significantly improved the baking quality.”(Line 451)

Other changes in the manuscript were also marked in red color.

We tried our best to improve the manuscript and hope that the correction will meet with approval.

Once again, thank you very much for your comments and suggestions.

Yours sincerely,

Jian Cai

Round 2

Reviewer 1 Report

Extensive changes have been made to the paper, especially in terms of improving the materials and methods section. I realize that the language is a problem, but there are still so many language errors in the text that need to be corrected. The ideal would be to make use of professional language editors. I have tried to make corrections as far as possible (please see attached pdf). 

Author Response

Dear ,

Thank you very much for giving us an opportunity to revise our manuscript again , we also appreciate your valuable comments on our manuscript. Those comments are very helpful for revising and improving our paper. We have revised the manuscript carefully. Revised parts are marked in red in the paper. The main corrections in the paper and the responses to your comments are as following:

 Point 1: Extensive changes have been made to the paper, especially in terms of improving the materials and methods section. I realize that the language is a problem, but there are still so many language errors in the text that need to be corrected. The ideal would be to make use of professional language editors. I have tried to make corrections as far as possible (please see attached pdf)  

 Response 1:Thanks for your advice, firstly, thank you very sincerely again for pointing out the mistakes for us. Secondly, according to your suggestions, the language problems have been revised in the manuscript one by one. Finally, thanks again for your advice and guidance.

Other changes in the manuscript were also marked in red color.

We tried our best to improve the manuscript and hope that the correction will meet with approval.

Once again, thank you very much for your comments and suggestions.

Yours sincerely,

Jian Cai

Reviewer 2 Report

Thanks for the corrections and the changes made.

Please find additional comments to improve your manuscript:

Line 10: to test the significant effects

Line 52-55:  Please, could you clarify this sentence? It is out of context and therefore incomprehensible

Line 110: oscillated is not the right term. "Stirred" would be more appropriate.

Line 110: Please express centrifugal force in g. Do the same thing  line 146.

Line 111: "repeat…". Please use the passive voice.

Lines 112-113: for NaCl and NaOH, add (w/v) or preferably indicate  molarity and normality

Line 120: I still do not understand your method. What is rinsed twice?

Line 120-122: Please rewrite using passive voice

Line 135: LMW-GS were

Line 136: "weighted…". Please rewrite using passive voice

Line 154: the proteins are immobilised in the gel

Line 158: please re-write the sentence "Quantitative…" and indicate the name of the equipment used

Line 168-171: The name of the method (with reference) and the name of the equipment are enough for describing GI measurement.

Line 208: prefer spike instead panicle to be in accordance with table 1

Line 210: CK is always used without explanation of the acronym

Lines 251-252: why do you use the verb to regulate and the word regulation? I do not understand it in this context.

Author Response

Dear,

Thank you very much for giving us an opportunity to revise our manuscript again , we also appreciate your valuable comments on our manuscript. Those comments are very helpful for revising and improving our paper. We have revised the manuscript carefully. Revised parts are marked in red in the paper. The main corrections in the paper and the responses to your comments are as following:

 Point 1: Line 10: to test the significant effects.

Response 1: Thanks for your advice, according to your suggestion, the word has been deleted in the manuscript.

Point 2: Line 52-55: Please, could you clarify this sentence? It is out of context and therefore incomprehensible

Response 2: Thanks for your advice, firstly, this section mainly quoted the effect of sulfur fertilizer on the quality of flour processing of previous studies. Secondly, “Addition of S increased loaf volume significantly at two sites where grain S concentration was also significantly increased and grain N:S ratio decreased.” was a reference to previous research results, because I did not express clearly, so it caused ambiguity. Lastly, it meant the sulfur increased the volume of bread, which suggested the effect of sulfur on the quality of bread varies with the variety and location of the bread. For the avoidance of ambiguity, it has been changed to “Previous studies suggested addition of S increased loaf volume significantly at two sites where grain S concentration was also significantly increased and grain N:S ratio decreased” in the manuscript. (Line 53)

Point 3: Line 110: oscillated is not the right term. "Stirred" would be more appropriate.

Response 3: Thanks for your advice, according to your suggestion, the word has been revised in the manuscript.

 Point 4: Line 110: Please express centrifugal force in g. Do the same thing line 146.

Response 4: Thanks for your advice, according to your suggestion, the word has been revised in the manuscript.

Point 5: Line 111: "repeat…". Please use the passive voice.

 Response 5: Thanks for your advice, according to your suggestion, the word has been revised in the manuscript.

 Point 6: Lines 112-113: for NaCl and NaOH, add (w/v) or preferably indicate molarity and normality in the manuscript.

Response 6: Thanks for your advice, according to your suggestion, it has been revised “The extracts of globulin, gliadin, and glutelin were obtained in 100 g L-1 NaCl, 70% (w/v) ethanol , and 2 g L-1 NaOH, respectively.” in the manuscript. (Line 115-116)

Point 7: Line 120: I still do not understand your method. What is rinsed twice?

Response 7: Thanks for your advice, firstly, “rinsed twice” meant rinsing tubes containing flour sample. Secondly, the aim was to remove the protein dissolved in the SDS solution to purify. Lastly, it has been revised ”The tubes with flour sample was rinsed 2 times with 2 ml SDS (1.5%) solution and was drained upside down.” in the manuscript. (Line 123)

Point 8: Line 120-122: Please rewrite using passive voice.

Response 8: Thanks for your advice, it has been revised “The tubes with flour sample was rinsed 2 times with 2 ml SDS (1.5%) solution and was drained upside down. Then 2 ml 0.2% sodium hydroxide solution was added, swirled and shook for 30 minutes, at last, 3 ml of biuret reagent was added, shook and rest for 30 min.” in the manuscript. (Line 123-126)

Point 9: Line 135: LMW-GS were

Response 9: Thanks for your advice, it has been revised in the manuscript.

Point 10: Line 136: "weighted…". Please rewrite using passive voice

Response 10: Thanks for your advice, it has been revised “To weigh 0.08 g flour, 1.5 mL n-propanol (solution A) was added…” in the manuscript. (Line 141-142)

Point 11: Line 154: the proteins are immobilised in the gel

Response 11: Thanks for your advice, I'm sorry for the misnomer. it has been revised After the electrophoresis, the gel was washed in 12% trichloroacetic acid for 10 min,” (Line 159)

Point 12: Line 158: please re-write the sentence "Quantitative…" and indicate the name of the equipment used

Response 12: Thanks for your advice, it has been revised “Quantitative analysis was done with Quantity One. (LICHEN, Changsha, China) (Line 163-164)

 Point 13: Line 168-171: The name of the method (with reference) and the name of the equipment are enough for describing GI measurement.

Response 13: Thanks for your advice, firstly it has been revised “Gluten value referred to the determination of wet gluten in wheat flour by mechanical means according to Chinese National Standard GB/T 5506.2-2008.” (Line 174-175). Secondly this method came from reference [26].

Point 14: Line 208: prefer spike instead panicle to be in accordance with table 1

Response 14: Thanks for your advice, I'm sorry for the misnomer. It has been revised in the manuscript.

Point 15: Line 210: CK is always used without explanation of the acronym

Response 15: Thanks for your advice, I'm sorry for the misnomer. It has been revised “Similarly, S(J) increased the spike number and yield above S0 by 12.5% and 7.8%, respectively.” in the manuscript. (Line 218).

Point 16: Lines 251-252: why do you use the verb to regulate and the word regulation? I do not understand it in this context.

Response 16: Thanks for your advice, I'm sorry for the misnomer. It has been revised “In addition, cysteine fertilizer showed better effects in increasing the contents of HMW-GS compared with sulfur fertilizer in two stages, but the increasing effect on the contents of LMW-GS was inconsistent.” (Line 258-261).

Other changes in the manuscript were also marked in red color.

We tried our best to improve the manuscript and hope that the correction will meet with approval.

Once again, thank you very much for your comments and suggestions.

Yours sincerely,

Jian Cai

Reviewer 3 Report

The authors made all requested changes, and the manuscript is suitable for publication in the present form.

Author Response

Dear ,

Thank you very much for your valuable comments on our manuscript.